# Deactivation and Regeneration of Nitrogen Doped Carbon Catalyst for Acetylene Hydrochlorination

**DOI:** 10.3390/molecules28030956

**Published:** 2023-01-18

**Authors:** Fangjie Lu, Qinqin Wang, Mingyuan Zhu, Bin Dai

**Affiliations:** 1School of Chemistry and Chemical Engineering, Shihezi University, Shihezi 832003, China; 2College of Chemistry and Chemical Engineering, Yantai University, Yantai 264010, China

**Keywords:** acetylene hydrochlorination, carbon deposition, carbon material, regeneration

## Abstract

The poor stability of carbon materials doped with nitrogen limited their development in acetylene hydrochlorination. Therefore, investigating the deactivation reasons of carbon catalysts and researching regeneration methods became the research focus. Herein, carbon-nitrogen materials were synthesized by one-step pyrolysis, which using biomass materials with high nitrogen content, the synthesized material was used in an acetylene hydrochlorination reaction. The acetylene conversion rate of D-GH-800 catalyst was up to 99%, but the catalytic activity decreased by 30% after 60 h reaction. Thermogravimetric analysis results showed that the coke content was 5.87%, resulting in catalyst deactivation. Temperature-programmed desorption verified that the deactivation was due to the strong adsorption and difficult desorption of acetylene by the D-GH-800 catalyst, resulting in the accumulation of acetylene on the catalyst surface to form carbon polymers and leading to the pore blockage phenomenon. Furthermore, based on the catalyst deactivation by carbon accumulation, we proposed a new idea of regeneration by ZnCl_2_ activation to eliminate carbon deposition in the pores of the deactivated catalyst. As a result, the activity of D-GH-800 was recovered, and lifetime was also extended. Our strategy illustrated the mechanism of carbon deposition, and the recoverability of the catalyst has promising applications.

## 1. Introduction

Acetylene hydrochlorination is a key step in the production of vinyl chloride monomers in coal-rich regions. However, the mercury-based catalysts are susceptible to loss, polluting the environment and harming human health [1,2]). Although precious metal catalysts (Au [3], Ru [4], and Pd [5]) show excellent catalytic activity, they are extremely expensive and have not yet been commercially used on a large scale.

During recent years, nitrogen-doped carbon materials have received much focus as potent catalysts for hydrochlorination of acetylene, whose catalytic activity was comparable to that of metal catalysts [6,7]. From our earlier work, we observed that the g-C_3_N_4_/AC catalyst prepared by loading graphite carbonitride (g-C_3_N_4_) on activated carbon (AC) exhibited a high acetylene conversion of 75%, identifying the key role of nitrogen atoms in carbon-nitrogen materials as adsorption sites for hydrogen chloride [8]. After that, researchers tried to explore efficient nitrogen-doped carbon catalysts for acetylene hydrochlorination. Wei et al. reported pyridinic N with full bonding as the most stable and conductive N state in N species [9]. Li et al. [10] speculated that the superior activity could be due to the adsorption and activation of the gases by highly electronegative heteroatoms and pyridinic N^+^O^−^ containing π-functional groups. Therefore, more researchers have increased the proportion of pyridinic-like nitrogen in N-enriched carbon materials and reported a variety of outstanding acetylene hydrochlorination catalysts [11]. The effect of nitrogen was also manifested in the formation of defective carbon rings, caused by the loss of N during the heat treatment of the prepared carbon-nitrogen materials, and other researchers have attempted to control defects in nitrogen-doped carbon catalysts [12]. Qiao et al. obtained nitrogen defects in g-C_3_N_4_ frameworks with porous structures, which greatly improved the adsorption of hydrogen chloride and acetylene, with acetylene conversions of 94.5% [13]. Researchers devoted to the catalytic performance of nitrogen-doped carbon materials have largely achieved the catalytic activity of metal catalysts, but the stability was still unsatisfactory for industrial applications.

The catalytic activity of nitrogen-doped carbon materials often decreased gradually with reaction time [14,15], but the deactivation reason was still unclear, which was crucial for further optimization of the catalyst. Firstly, Bao et al. [16] investigated the deactivation reason of nitrogen-doped carbon catalysts and concluded that the deactivation was caused by the carbon-like sediments generated in the catalytic reaction during the exothermic reaction, which prevented the reactants from accessing the active sites, resulting in a decrease in activity. Secondly, Dong et al. [17] attributed the deactivation to the loss of some of the active component, nitrogen-containing diheterocycle molecules [DBU][Cl], leading to the deactivation. For different causes of catalyst deactivation, the existing methods of NH_3_ regeneration can increase the nitrogen content but cannot change the specific surface area of the catalyst and do not effectively solve the carbon accumulation problem [18,19]. Therefore, we believe that there is an urgent need to explore the real cause of the catalyst deactivation and to find a new idea for regeneration that can restore catalytic activity.

In this paper, we successfully prepared a series of carbon and nitrogen materials by using D(+)-glucosamine hydrochloride (D-GH), a cheap biomass material with high nitrogen content [20], as the precursor, and regulating the N content in the catalyst by direct pyrolysis. The D-GH-800 catalyst with high-density pyridine N exhibited excellent acetylene conversion. Thermogravimetric, specific surface area and X-ray photoelectron spectroscopy analysis were used to investigate the N content and carbon deposition of the deactivated catalysts to find the real cause of the catalyst deactivation. Furthermore, based on the deactivation cause, we invoked the role of ZnCl_2_ activation in the preparation of porous structures [21,22], proposed a new regeneration method, and the activity of D-GH-800 was recovered; furthermore, the lifetime was extended observably. In short, we successfully synthesized a catalyst with excellent acetylene conversion and effective recyclability in acetylene hydrochlorination.

## 2. Results and Discussion

### 2.1. Catalyst Performance Evaluation

We successfully prepared a series of carbon and nitrogen materials by simple pyrolysis of four nitrogen-containing glucose biomass materials selected as raw materials for the preparation of catalysts, and we evaluated their catalytic performance, as shown in Figure 1a. The D-GH catalysts showed optimum catalytic activity. Since only high-temperature pyrolysis was used throughout the catalytic process, and considering that the calcination atmosphere and the pyrolysis temperature can greatly influence the material structure of the catalyst, we investigated two conditions with the optimal D-GH catalyst as an example. The catalysts under different calcination atmospheres showed variability in acetylene conversion, with the catalysts prepared under an argon atmosphere having the highest acetylene conversion (Appendix A). Figure 1b shows the results of the catalyst performance evaluation at different calcination temperatures, and the optimum calcination temperature was 800 °C, with acetylene conversion of 96%. Compared to the reported catalyst TOFs data, D-GH-800 showed the largest TOF values, comparing favorably with most of the previous studies (Appendix A). Therefore, we selected D(+)-glucosamine hydrochloride as the optimal raw material for the successful preparation of biomass carbon and nitrogen materials, which showed the optimal catalytic activity under the calcination atmosphere of Ar and the calcination temperature of 800 °C, achieving the desired goal through a simple catalyst synthesis route in order to provide a convenient condition for subsequent catalyst regeneration.

### 2.2. Physical Characterization

The surface morphology of the D-GH samples was explored by SEM, as shown in Figure 2a,b. D-GH-800 have a multi-layered porous structure with more pore structures than D-GH-600, and it can be determined that increasing the pyrolysis temperature appropriately led to an increase in the number of voids in the D-GH material. As the pyrolysis temperature increased to 1000 °C (Figure 2c,d), the pore structure changed significantly, and excessive pyrolysis led to an increase in pore size and carbon structure buildup, resulting in the collapse and blockage of the pore structure. This conclusion can be verified by the catalyst BET data in Table 1. When the pyrolysis temperature reaches 800 °C, the specific surface area and pore volume of the D-GH material gradually increases and the pore size decreases, probably due to the gaseous molecules (NO, NO_2_, CO_2_) formed during the rising temperature increasing the pore structure of the catalyst [23]. However, when the temperature >800 °C, the high temperature led to the collapse of the pore structure, and the specific surface area started to decrease. D-GH-800 had a larger BET-specific surface area and pore volume.

The TEM image of D-GH-800 shows the loose structure of the catalyst, with an irregular structure around the edges (Figure 2e), and elemental mapping analysis (Figure 2f–i) showed that the three elements C, O, and N were evenly distributed within the catalyst, indicating the presence of many dispersed, active, center N elements in the material [24]. The loss of gaseous molecules during pyrolysis was explored by XPS elemental content analysis. The relative content of N and O gradually decreased, and the content of C increased as the calcination conditions increased, and the XRD results also indicated an increase in the degree of graphitization of the carbon material (Appendix A). The extent of the defects in the catalyst increased as the N content decreased rapidly (Appendix A). We deduced that the D-GH-800 catalyst exhibited excellent catalytic activity since it has the largest surface area, optimal structure, suitable N content, and defect structure, but that the D-GH-600 catalyst was slightly lower than that of D-GH-800, simply because it has the highest N content.

### 2.3. Analysis of Catalytic Active Sites

In order to investigate the active role of nitrogen in the catalytic process and to determine the number of central elements that act as active components, N1s XPS analysis was performed on the D-GH catalyst. As shown in Figure 3 and Appendix A, there was a coexistence of pyridinic N, pyrrolic N, graphitic N, and oxidized N in the D-GH catalyst [7,25]. It was noteworthy from the detailed data of the nitrogen species content that pyridinic N showed a trend of increasing and then decreasing, while graphitic N and oxidized N gradually increased. Combined with the catalytic performance of the D-GH catalyst (Figure 1b), the trend of the pyridinic N content in the catalyst was consistent with the change in the acetylene conversion, with the D-GH-800 catalyst having the optimum catalytic performance, with a high pyridinic N content of 34.64%, suggesting that pyridinic N was the active site for the reaction, a finding that was consistent with our previous results. In conclusion, in combination with the multiple characterization results, the D-GH-800 catalyst had the largest surface area, the active site of the highest amount of pyridinic N, and the highest catalytic activity for acetylene hydrochlorination.

### 2.4. Stability and Deactivation Reasons

Next, the stability of the best catalytic D-GH-800 catalyst was investigated. By screening the reaction conditions of the optimum catalyst, we chose to determine the stability at the reaction temperature of 220 °C and the reaction airspeed of 50 h^−1^ (Appendix A). From Figure 4a, after 50 h, the catalytic activity of the D-GH-800 catalyst decreased from 99% to 73%, with a significant deactivation. We explored the possible causes of catalyst deactivation to provide a clear direction for subsequent regeneration experiments. It should be noted that the results of the N1s XPS analysis showed no significant changes in the total N content and major active-site pyridine N content before and after the catalyst reaction (Figure 4b, Appendix A), which is not sufficient to explain the deactivation of the D-GH-800 catalyst. However, Air-TG testing of the catalyst before and after the reaction from Figure 4c indicated a coke content of approximately 5.87%, a finding that contrasts with the previously reported high carbon deposition of 5.52% [26] and 6.30% [9]. Meanwhile, the specific surface area of the catalyst decreased significantly after the reaction (Figure 4d, Appendix A), the pore volume decreased, and the pore size increased, so carbon deposition could explain the deactivation of the D-GH-800 catalyst. In addition, SEM images of the D-GH-800 catalyst after the reaction observed that most of the pores were plugged after the reaction, concentrated on the macropores. (Appendix A). These observations suggested a significant reduction in catalyst activity due to the blockage of pores by the carbon deposition generated on the catalyst, independent of the number of active sites.

Further investigation into the mechanism of carbon deposition generation on the D-GH-800 catalyst is explained by TPD analysis (Appendix A). According to the study of the key role of nitrogen atoms in carbon and nitrogen materials, HCl was adsorbed on the surface of D-GH-800 through N [8,27], while C_2_H_2_ was adsorbed on the C atoms of the catalysts due to the interaction with C. According to the TPD analysis, the desorption temperature of HCl on the D-GH catalyst was essentially the same, and the desorption area decreased with increasing calcination temperature. Combined with the effect of nitrogen, the adsorption of HCl by the catalyst gradually increased as the nitrogen content increased, indicating that the maximum adsorption of HCl by the D-GH-600 catalyst was attributed to having the maximum N content. As shown in Figure 4d, the D-GH-800 catalyst exhibited the maximum C_2_H_2_ adsorption and the highest desorption temperature. We speculate that although the D-GH-800 catalyst has a high catalytic performance for a short time, due to the low HCl adsorption and difficult desorption of C_2_H_2_, the excess adsorbed C_2_H_2_ tends to accumulate on the catalyst surface and form carbon macromolecules, resulting in carbon plugging. In summary, the D-GH-800 catalyst produces carbon deposits due to the strong adsorption and difficult desorption of C_2_H_2_ and the accumulation of carbon macromolecules, which eventually leads to the deactivation of the catalyst and poor stability.

### 2.5. Stability and Deactivation Reasons

The challenge was to effectively remove surface carbon deposition without damaging its intrinsic carbon network, especially the active sites within the network. We proposed a novel ZnCl_2_ regeneration method for our investigation of carbon deposition deactivation of D-GH-800 catalysts. We changed the conventional thinking that ZnCl_2_ was applied to prepare porous catalysts. We improved the utilization rate of the D-GH-800 catalyst by treating the regenerated lifetime D-GH-800 catalyst (Used) with ZnCl_2_ at a high temperature. By designing experiments, Used: ZnCl_2_ = 1:0.5 and calcination temperature of 800 °C was the optimal regeneration experimental condition, and the catalytic activity of 0.5ZnCl_2_/Used-800 recovered to 84.28% (Figure 5a and Appendix A), while the Used-800 catalyst without ZnCl_2_ did not recover the initial catalytic activity, indicating that the ZnCl_2_ regeneration method can effectively achieve reuse of the catalyst. We found that the mass reduction of the regenerated catalyst could be attributed to the removal of carbon accumulation in the pores by the effect of ZnCl_2_. This conclusion was also confirmed by the BET data, as seen in Figure 5c and Appendix A, where the surface area of the regenerated catalyst increased significantly from 14.24 m^2^g^−1^ to 507.55 m^2^g^−1^. Importantly, XPS showed no significant change in the N contents and active-site pyridine N content of the catalyst before and after regeneration (Figure 5d and Appendix A). After the reaction, 0.5ZnCl_2_/Used-800used had a reduced specific surface area, and the surface structure was destroyed (Appendix A, Table 1). These observations suggested that ZnCl_2_ at a high temperature removed a large amount of carbon deposition from the catalyst surface without changing the number of active sites. The effective removal of carbon deposition restores the surface area and pore structure of the catalyst and improves the utilization of the catalyst. Therefore, it was feasible to remove the carbon deposition by the ZnCl_2_ regeneration method.

## 3. Materials and Methods

### 3.1. Materials

D(+)-Glucosamine hydrochloride, D-Glucosaminic, and D-Glucosamine sulphate were from Aladdin Co., Ltd., Shanghai, China. 2-Acetamido-2-deoxy-L-glucopyranose, N-Methyl-D-glucamine, and Zinc chloride were purchases from MACKLIN Co., Ltd., Shanghai, China. Concentrated hydrochloric acid (HCl, 99.0%) was obtained from Sinopharm Chemical Reagent Co., Ltd., Shanghai, China.

### 3.2. Catalyst Preparation

#### 3.2.1. Preparation of Series Carbon and Nitrogen Materials

Several nitrogen-containing glucose biomass materials (D-Glucosaminic (D-G), D(+)-Glucosamine hydrochloride (D-GH), D-Glucosamine sulphate (D-GS), 2-Acetamido-2-deoxy-L-glucopyranose (NA-DH), and N-Methyl-glucose (NM-DG)) were used as raw materials for the preparation of the catalysts in quartz boats, which were heated at a rate of 5 °C/min to 300 °C for 1 h at a flow rate of 60 mL/min^−1^ and then ramped up to 800 °C for 2 h under Ar. The range of carbon and nitrogen materials was successfully prepared. The D-GH-A (A = N_2_, Ar, NH_3_ and H_2_/Ar) samples were prepared in the same way as the above pyrolysis method, except that the chosen calcination atmosphere was set to N_2_, Ar, NH_3_, and 5% H_2_/Ar, respectively.

#### 3.2.2. Synthesis Procedure for D-GH-T Samples

D(+)-glucosamine hydrochloride (10 g) was placed as a precursor in a quartz vessel in an argon tube furnace with an argon flow rate of 60 mL min^−1^. The first stage of heating was set to a final temperature of 300 °C, with a rate of 5 °C/min and a holding time of 1 h. Then, a second stage of heating was carried out on this basis, setting the final temperature to 800 °C with a rate of 5 °C/min and a holding time of 2 h. The solid was allowed to cool to room temperature and, finally, about 2.35 g of solid D-GH-800 catalyst was obtained. The other D-GH-T (T = 600, 1000, 1200 °C) samples were prepared in the same way as D-GH-800, except that the final temperatures of the second stage were set to 600 °C, 1000 °C, and 1200 °C, respectively.

#### 3.2.3. Synthesis of Regeneration xZnCl_2_/Used-H Catalysts

First, zinc chloride (1 g) was dissolved in 30 mL of water and stirred uniformly for 30 min. Then, 2 g of D-GH-800 catalyst after the life test (catalyst abbreviated as Used) was prepared, added to the zinc chloride solution, stirred at 110 °C for 6 h, and then dried. The dried precursor was heated to 800 °C under Ar, 5 °C/min, 2 h. The calcined sample was washed with 2 M HCl solution for 10 h and dried at 80 °C for 24 h to obtain 0.5ZnCl_2_/Used-800 as about 1.8 g. As a comparison, xZnCl_2_/Used-800 (x = 0.25, 1 and 2) was prepared by adjusting the mass ratio of Used to ZnCl_2_ 1:0.25, 1:1, and 1:2. The calcination temperatures of the 0.5ZnCl_2_/Used catalysts were varied to 600 °C, 1000 °C, and 1200 °C, and the catalysts were named 0.5ZnCl_2_/Used-H (H = 600, 1000, and 1200 °C). Meanwhile, the Used catalyst was prepared using the same pyrolysis procedure without the addition of ZnCl_2_ to produce the Used-800 catalyst as a comparison.

### 3.3. Characterizations

X-ray diffraction analysis (XRD) data were achieved by scanning from 10° to 90° using a D8 VENTURE/QUEST. Thermogravimetric analysis (TGA) of the catalysts was obtained by heating to 800 °C in an air atmosphere at 10 °C/min on a NETZSCH STA 449 F5 instrument. Scanning electron microscopy (SEM) (Hitachi S4800) was used to analyze the morphology of the samples. X-ray photoelectron spectroscopy (XPS) was used for the determination of elemental content of catalysts exposed to air for long times. Temperature programmed desorption (TPD) was analyzed by desorption using a Chem BET Pulsar TPD in a helium atmosphere at 10 °C/min. Analytical samples Brunauer–Emmett–Teller (BET) were passed through the Micromeritics ASAP 2020 instrument. Inductively coupled plasma optical emission spectrometry (ICP-OES) analysis (ICAP7400) was performed to test for Zn atomic content. The structure of catalyst was analyzed by transmission electron microscopy (TEM, FEI Talos F200X). Analyzed the distribution of O, N, and C elements by elemental mapping to assist in phase identification or structural analysis.

### 3.4. Description of Catalytic Tests and Analytical Criteria

The catalytic effect of the fixed-bed reactor was measured with a catalyst dosage of 1.2 g (2 mL). Nitrogen was introduced to remove air from the tube and heated to 180 °C at 5 °C/min under N_2_ atmosphere. The test was started by passing in acetylene and hydrogen chloride reaction gas (C_2_H_2_ = 36 h^−1^, (V_HCl_/V_C2H2_ = 1.15). The unreacted hydrogen chloride was absorbed by a solution containing sodium hydroxide. Simultaneous gas chromatography (GC) was performed to detect the reaction product content online. The conversion of acetylene (X_A_) is used as a standard for catalytic performance. ΦA0, ΦA is the volume fraction of C_2_H_2_ before and after the reaction.
(1)$$X_{A}=\frac{\Phi A0-\Phi A}{\Phi A0}\cdot 100\%$$

## 4. Conclusions

In summary, we successfully proposed a deactivation cause and a novel regeneration method for nitrogen-doped carbon materials in an acetylene hydrochlorination reaction. The acetylene conversion of D-GH-800 catalyst with 6.28% nitrogen content synthesized by one-step pyrolysis of biomass materials was up to 99%. Our designing experiments showed that the catalytic activity of the catalyst was closely associated with the density of the pyridinic N content. The characterization results showed that the deactivation of D-GH-800 catalyst was due to the strong adsorption and difficult desorption of acetylene, resulting in an accumulation of acetylene on the catalyst surface forming carbon polymers and leading to the pore blockage phenomenon. Furthermore, based on the deactivation cause, an effective ZnCl_2_ regeneration method was proposed, and the carbon deposition in the pores of the deactivated catalyst was eliminated, improving the utilization of the D-GH-800 catalyst. This research clarified that the causes of carbon and nitrogen catalyst deactivation played an important role in improving catalytic stability and provides a promising regeneration method.

## Figures and Tables

**Figure 1 molecules-28-00956-f001:**
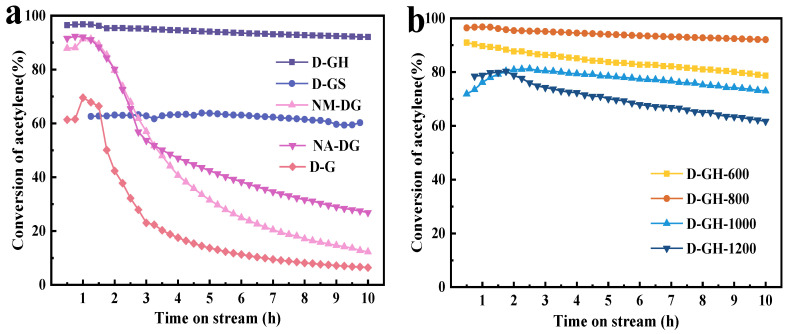
(**a**) The catalytic performance of carbon and nitrogen materials prepared from different biomass materials. (**b**) The catalytic performance of D-GH catalysts. Reaction conditions: T = 180 °C, C_2_H_2_ = 36 h^−1^, velocity V_HCl_/V_C2H2_ = 1.15.

**Figure 2 molecules-28-00956-f002:**
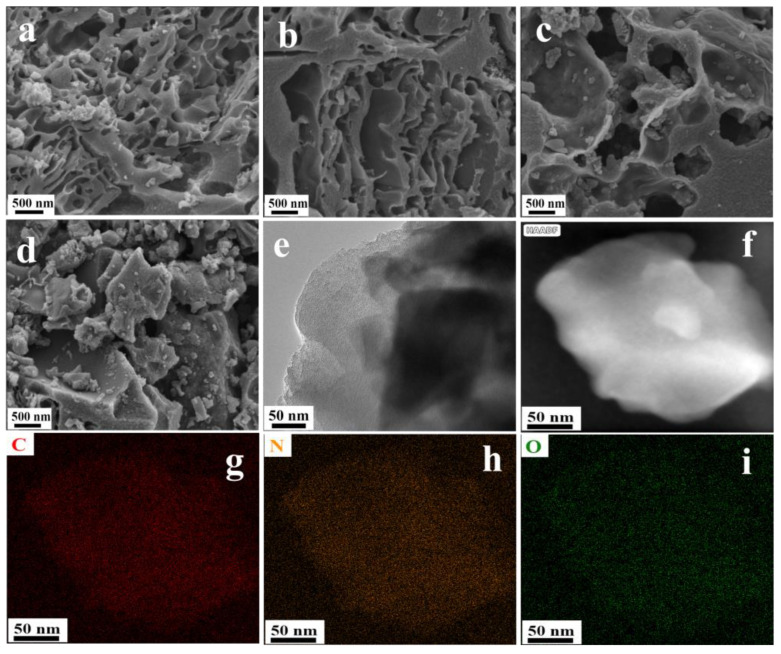
SEM image of D-GH catalysts; (**a**–**d**) stands for D-GH-600, D-GH-800, D-GH-1000, and D-GH-1200, respectively. (**e**) TEM image of D-GH-800 catalyst. (**f**–**i**) HAADF-STEM mapping (C, N, and O) of D-GH-800.

**Figure 3 molecules-28-00956-f003:**
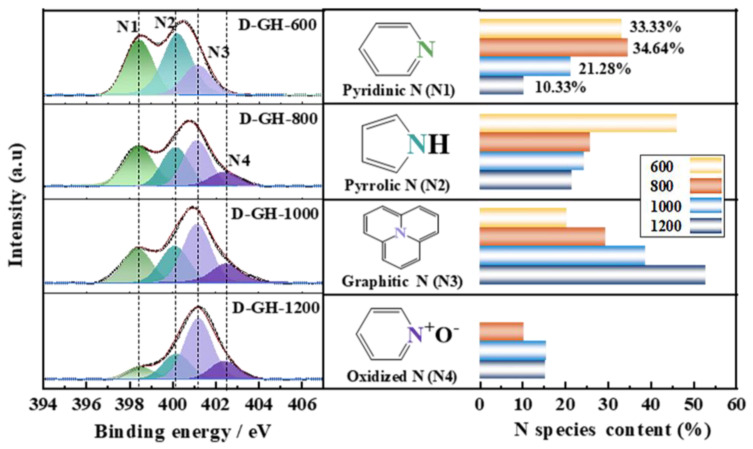
N 1s XPS spectra of D-GH-T and the pyridinic-N, pyrrolic-N, graphitic-N, and oxidized N content on N 1s XPS spectra.

**Figure 4 molecules-28-00956-f004:**
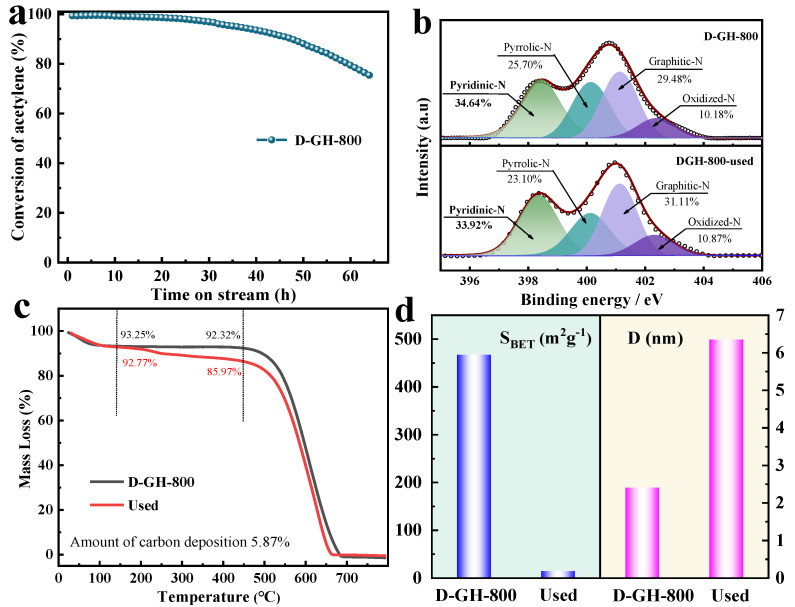
(**a**) The catalytic stability of D-GH-800. Reaction conditions: T = 220 °C, C_2_H_2_ = 50 h^−1^, V_HCl_/V_C2H2_ = 1.15. (**b**) High-resolution XPS N1s spectra for D-GH-800 and Used. (**c**) Air-TG curves of the unreacted and reacted D-GH-800 recorded under air atmosphere. (**d**) The specific surface area data of Used and 0.5ZnCl_2_/Used-800 catalysts.

**Figure 5 molecules-28-00956-f005:**
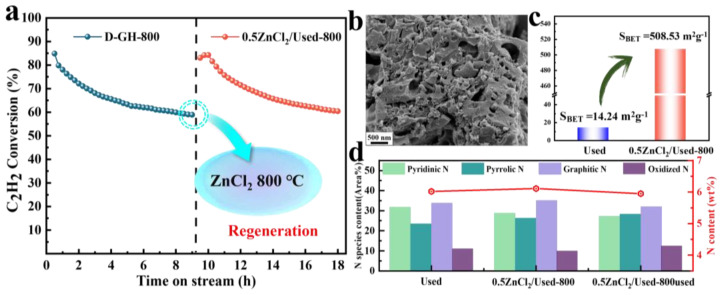
(**a**) The catalytic stability of D-GH-800 and 0.5ZnCl_2_/Used-800. Reaction conditions: T = 180 °C, C_2_H_2_ = 180 h^−1^, V_HCl_/V_C2H2_ = 1.15. (**b**) SEM image of 0.5ZnCl_2_/Used-800 catalysts. (**c**) The specific surface area data of Used and 0.5ZnCl_2_/Used-800 catalysts. (**d**) The pyridinic-N, pyrrolic-N, graphitic-N, and oxidized N content on N 1s XPS spectra of D-GH-800, 0.5ZnCl_2_/Used-800 and 0.5ZnCl_2_/Used-800 catalysts.

**Table 1 molecules-28-00956-t001:** Pore structure parameters and element content of various catalysts.

Catalyst	S_BET_ (m^2^g^−1^)	V (cm^3^g^−1^)	D (nm)	C	N	O
D-GH-600	37.40	0.05	5.27	84.84	8.52	6.63
D-GH-800	466.99	0.25	2.49	87.73	6.28	5.99
D-GH-1000	368.51	0.24	2.41	90.04	4.08	5.88
D-GH-1200	50.89	0.02	2.38	92.56	1.96	5.48

## Data Availability

The data presented in this study are available on request from the corresponding author.

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
