# Peer review of "Deactivation and Regeneration of Nitrogen Doped Carbon Catalyst for Acetylene Hydrochlorination"

_molecules, 2023, doi:10.3390/molecules28030956_

Round 1

Reviewer 1 Report

The manuscript by Lu et al. discussed the deactivation of nitrogen doped carbon catalysts for acetylene hydrochlorination, and proposed a new regeneration method by ZnCl2 activation. The viewpoints of this manuscript are pretty clear and of novelty, which would be benefiting for improving the stability of nitrogen doped carbon catalysts. However, the following issues still should be addressed before its publication.

(1) The experimental requires more specific detail to allow reproductions of the result. In Figure 1a, calcination atmosphere is missing in the heat treatment.

(2) When pyrolysis temperature is above 800 °C, the specific surface area of the D-GH material gradually increased, the pore volume increased. This results are conflicting with the data in Table 1. Please explain it.

(3) Why the content of nitrogen and pyridine N remained unchanged in the used catalysts, while the catalytic performance changed significantly.

(4) Figure 5d show that the pyridine nitrogen decreases, and the pyrrole nitrogen increases after performing the ZnCl2 treatment. Did the catalyst regeneration restore the catalytic activity?

(5)  The authors proposed the use of ZnCl2 to treat the used catalyst at higher temperatures, but no comparative experiments without the addition of ZnCl2 were performed.

(6) The specific role of ZnCl2 is not clarified in the manuscript. Is it to eliminate carbon deposition or to increase the pore structure? The authors need to give a detailed explanation.

(7) There are several formal errors in this draft. For example, “D-GH-800 had a multi-layered porous structure with more pore structures than D-GH-600”; there is an error in the explanation of Table 1, etc. The authors should double check.

Author Response

Reviewer #1:

We sincerely thank for your valuable feedback that we have used to improve the quality of our manuscript. The reviewer comments are laid out below in italicized font and specific concerns have been numbered. Our response is given in normal font and changes/additions to the manuscript are given in the red text.

Question 1 The experimental requires more specific detail to allow reproductions of the result. In Figure 1a, calcination atmosphere is missing in the heat treatment.

Response: According to the Reviewer’s suggestion, we re-examine this issue. We have made changes in the corresponding sections of the manuscript.

Corresponding part can be found in lines 235-237.

Question 2 When pyrolysis temperature is above 800 °C, the specific surface area of the D-GH material gradually increased, the pore volume increased. This results are conflicting with the data in Table 1. Please explain it.

Response: We sincerely thank the reviewer for careful reading. We have corrected the “pyrolysis temperature > 800 °C, the specific surface area of the D-GH material gradually increased, the pore volume increased” into “When the pyrolysis temperature reaches 800°C, the specific surface area and pore volume of the D-GH material gradually increase, and the pore size decreases.”

Corresponding part can be found in lines 114-117.

Question 3 Why the content of nitrogen and pyridine N remained unchanged in the used catalysts, while the catalytic performance changed significantly.

Response: In this word, we found no significant changes in the total N content and the pyridine N content of the main active site before and after the reaction of the catalyst (Figure 4b, Table S2), and the Air-TG test indicated that the coke content was about 5.87%, while the specific surface area of the catalyst obviously decreased, and the pore volume decreased after the reaction (Figure S4, Table S2). These observations suggested a significant reduction in catalyst activity due to the blockage of pores by carbon deposition generated on the catalyst, independent of the number of active sites.

Question 4 Figure 5d show that the pyridine nitrogen decreases, and the pyrrole nitrogen increases after performing the ZnCl2 treatment. Did the catalyst regeneration restore the catalytic activity?

Response: After performing the ZnCl2 treatment, the catalyst recovered its catalytic activity. We found that the regeneration was achieved mainly because the action of ZnCl2 at high temperature effectively removed a large amount of carbon accumulation in the pores of the catalyst surface, restoring the surface area and pore structure of the catalyst. In contrast, the changes of pyridine nitrogen and pyrrole nitrogen played a smaller role in the restoration of activity and was not a major factor.

Question 5 The authors proposed the use of ZnCl2 to treat the used catalyst at higher temperatures, but no comparative experiments without the addition of ZnCl2 were performed.

Response: According to the Reviewer’s suggestion, we have supplemented the corresponding section in the manuscript.

The catalytic activity of 0.5ZnCl2/Used-800 recovered to 84.28% (Figure S6), while the Used-800 catalyst without ZnCl2 did not recover the initial catalytic activity, indicating that the ZnCl2 regeneration method can effectively achieve reuse of the catalyst.

Corresponding part can be found in lines 208-211, 261-262.

Question 6 The specific role of ZnCl2 is not clarified in the manuscript. Is it to eliminate carbon deposition or to increase the pore structure? The authors need to give a detailed explanation.

Response: We redefined the specific role of ZnCl2 according to the Reviewer's suggestion. The results of the yield, morphology and specific surface area of the regenerated 0.5ZnCl2/Used-800 catalyst indicate that the action of ZnCl2 at high temperatures removes a significant amount of carbon deposition from the pores of the catalyst surface without changing the number of active sites. The effective removal of carbon deposition restored the surface area and pore structure of the catalyst and improved the catalyst utilization. Therefore, the removal of carbon deposition by ZnCl2 regeneration is feasible.

Corresponding part can be found in lines 211-213, 219-223.

Question 7 There are several formal errors in this draft. For example, “D-GH-800 had a multi-layered porous structure with more pore structures than D-GH-600”; there is an error in the explanation of Table 1, etc. The authors should double check.

Response: We feel sorry for our carelessness. In our resubmitted manuscript, the grammar errors are revised. Thanks for your correction

Corresponding part can be found in lines 108-109, 121.

Reviewer 2 Report

The authors propose an interesting approach to prepare and regenerate biomass-based N-doped carbon materials towards acetylene hydrochlorination. The manuscript is well written and organized with complete dataset. I am pleased to recommend for publication after a minor revision. 

1/ The nitrogen free and nitrogen contained glucoses should be separated. It is very confusing when seeing D-glucose and D-GH or D-GS among nitrogen containing glucose materials. 

2/ Could you provide the pyrolysis yields ? 

Author Response

Reviewer #2:

According to the associate editor and reviewers' comments, we have made extensive modifications to our manuscript and supplemented extra data to make our results convincing.

Question 1 The nitrogen free and nitrogen contained glucoses should be separated. It is very confusing when seeing D-glucose and D-GH or D-GS among nitrogen containing glucose materials.

Response: It is a huge mistake for the overall quality of our article that the drug name was incorrect. We feel sorry for our carelessness. We have corrected it and we also feel great thanks for your point out.

In the study, several nitrogen-containing glucose biomass materials (D-Glucosaminic (D-G), D(+)-Glucosamine hydrochloride (D-GH), D-Glucosamine sulphate (D-GS), 2-Acetamido-2-deoxy-L-glucopyranose (NA-DH) and N-Methyl-glucose (NM-DG)) were prepared as raw materials to prepare the catalysts.

Corresponding part can be found in lines 232-237.

Question 2 Could you provide the pyrolysis yields?

Response: Thanks for the Reviewer’s suggestion, D-GH-800 catalyst has a pyrolysis yield of about 23.5% (10g of raw material harvested for about 2.35g of catalyst)

Corresponding part can be found in lines 246-248, 255-257.

Reviewer 3 Report

Article titled as “Deactivation and regeneration of nitrogen doped carbon catalyst for acetylene hydrochlorination” is interesting but needs following modifications.

1.     In abstract section authors should mention some important outcomes of present research along with numerical results data.

2.     In line 32 “C3N4/AC catalysts exhibited” what is AC, authors should add full name of the abbreviation where its first appears.

3.     Line 39, “Therefore, many research have effectively increased” authors make its correction according to the grammar.

4.     Line 78 “Zinc chloridewere purchases” there should be space between chloride and were. Also grammatical errors should be removed.

5.     Authors should add loading amount catalyst and acetylene conversion in conclusion section.

6.     Authors should add a comparison with related published work.

Author Response

Reviewer #3:

Thank you for your nice comments on our article. According to your suggestions, we have supplement several data here and corrected several mistakes in our previous draft. Based on your comments, we also attached a point-by-point letter to you and the other reviewers. We have made extensive revisions to our previous draft. The detailed point-by-point responses are listed below.

Question 1 In abstract section authors should mention some important outcomes of present research along with numerical results data.

Response: According to the Reviewer’s suggestion, we re-examine this issue. We have made changes in the abstract section of the manuscript.

Corresponding part can be found in lines 12-15

Question 2 In line 32“C3N4/AC catalysts exhibited” what is AC, authors should add full name of the abbreviation where its first appears.

Response: Thank you for pointing this out. We have corrected it to " “g-C3N4/AC catalyst prepared by loading graphite carbonitride (g-C3N4) on activated carbon (AC)”.

Corresponding part can be found in lines 34-35

Question 3 Line 39, “Therefore, many research have effectively increased” authors make its correction according to the grammar.

Response: Thanks for your careful checks. We are sorry for our carelessness. Based on your comments, we have made the corrections to make the word harmonized within the whole manuscript.

Corresponding part can be found in lines 41-44

Question 4 Line 78 “Zinc chloridewere purchases” there should be space between chloride and were. Also grammatical errors should be removed.

Response: We sincerely thank the reviewer for careful reading. As suggested by the reviewer, we have corrected the “Zinc chloridewere purchases” into “Zinc chloride were purchases”.

Corresponding part can be found in lines 228

Question 5 Authors should add loading amount catalyst and acetylene conversion in conclusion section.

Response: Thank you for pointing this out. The reviewer is correct, and we have revised conclusion section.

Corresponding part can be found in lines 289-290.

Question 6 Authors should add a comparison with related published work.

Response: We think this is an excellent suggestion. We think this is a good suggestion. We have added TOFs values for comparison with relevant published catalysts, which can be found in the revised manuscript.

Compared to the reported catalyst TOFs data, D-GH-800 showed the largest TOF values, comparing favorably with most previous studies (Table S1).

Corresponding part can be found in lines 95-97.

Round 2

Reviewer 3 Report

Authors have updated their draft according to the given suggestions so it maybe accepted in its present form.